# *Values, Ethics, Morals?*
# On the Use of Moral Concepts in NLP Research

**Karina Vida**
Data Science Group
University of Hamburg,
Germany

**Judith Simon**
Ethics in Information Technology
University of Hamburg,
Germany

**Anne Lauscher**
Data Science Group
University of Hamburg,
Germany

`{karina.vida, judith.simon, anne.lauscher}@uni-hamburg.de`

## Abstract

With language technology increasingly affecting individuals' lives, many recent works have investigated the ethical aspects of NLP. Among other topics, researchers focused on the notion of *morality*, investigating, for example, which moral judgements language models make. However, there has been little to no discussion of the terminology and the theories underpinning those efforts and their implications. This lack is highly problematic, as it hides the works' underlying assumptions and hinders a thorough and targeted scientific debate of morality in NLP. In this work, we address this research gap by (a) providing an overview of some important ethical concepts stemming from philosophy and (b) systematically surveying the existing literature on moral NLP w.r.t. their philosophical foundation, terminology, and data basis. For instance, we analyse what ethical theory an approach is based on, how this decision is justified, and what implications it entails. Our findings surveying 92 papers show that, for instance, most papers neither provide a clear definition of the terms they use nor adhere to definitions from philosophy. Finally, (c) we give three recommendations for future research in the field. We hope our work will lead to a more informed, careful, and sound discussion of morality in language technology.

## 1 Introduction

With Natural Language Processing (NLP) receiving widespread attention in various domains, including healthcare (e.g., Krahmer et al., 2022; Ji et al., 2022), education (e.g., Alhawiti, 2014; Srivastava and Goodman, 2021), and social media (e.g., Wang et al., 2019; Syed et al., 2019), the ethical aspects and the social impact of language technology have become more and more important (Hovy and Spruit, 2016; Blodgett et al., 2020; Weidinger et al., 2021).

In this context, recent research focused on the notion of *morality* (e.g., Araque et al., 2020;

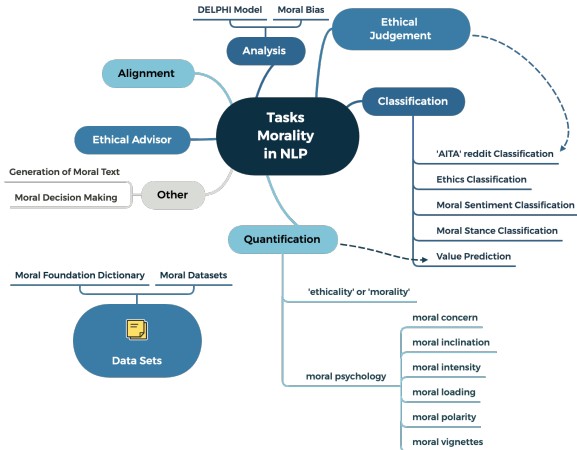

Figure 1: The landscape of problems tackled under the umbrella of *'morality in NLP'* and their connections. The dashed arrows indicate a connection between categories, which cannot be clearly distinguished.

Hendrycks et al., 2020; Hämmerl et al., 2022b, *inter alia*), for instance, with the goal of extracting morality and moral values automatically from text. The existing research landscape is manifold (cf. Figure 1), ranging, for example, from creating suitable data sets (e.g., Forbes et al., 2020; Sap et al., 2020), over investigating moral consistency in different languages (e.g., Hämmerl et al., 2022b) to constructing NLP models that are able to make moral judgements about input sentences (e.g., Shen et al., 2022; Alhassan et al., 2022).

Such attempts have also sparked controversy in the research community and the public media. As a widely discussed[1] example, DELPHI (Jiang et al., 2021a), has been criticised, among other reasons, for the *normative* nature of its judgements given the authors' goal of creating a model of *descriptive ethics* (Talat et al., 2022). We argue that this mismatch relates to a bigger problem in our community: **a lack of clear definitions coupled with a**

---

[1]E.g., coverage in New York Times: `https://www.nytimes.com/2021/11/19/technology/can-a-machine-learn-morality.html`

**confusion about important underlying concepts stemming from philosophy, psychology, and beyond**. As a result, it is unclear to what extent the foundations of moral philosophy can even be found in NLP research on morality and whether and how researchers are considering ethical theories and related philosophical concepts in NLP. In-depth knowledge on how NLP is dealing with the different shades of morality is missing – hindering a targeted scientific discussion in the field.

**Contributions.** We address this gap by surveying the state of research on the interplay of morality and NLP. Concretely, (**1**) we analyse 92 publications on morality in NLP resulting in the only survey on this topic to-date. We draw a map of the existing NLP tasks dealing with morality (e.g., classification of moral values), analyse the moral (i.e., philosophical and/or psychological) foundations pertaining to current research, and examine the existing data sets. To this end, (**2**) we are the first to provide a thorough overview of the most important concepts relating to morality for NLP. For instance, we discuss the different branches of moral philosophy and three main families of ethical theories (consequentialism, deontological ethics, and virtue ethics) to clarify common misconceptions in the community. We find, for instance, that (a) most papers do not refer to ethical principles, that (b) relevant philosophical terms (e.g., *'morality'*, *'ethics'*, and *'value'*) are often used interchangeably; and that (c) clarifying definitions of the terms used are rarely provided. Finally, (**3**) we use our insights to provide three recommendations for future research.

## 2 Background and Terminology

*Ethics* has undeniably become a critical topic within NLP.[2] However, as we show in §4, the term is often used without specification, leaving ambiguity about what branch of moral philosophy authors refer to. Here, we introduce the precise terminology we will use in the remainder of this work.

**Ethics.** The branch of philosophy that deals with human practice, i.e., human actions and their evaluation, is called ethics. Ethics is composed of four branches, each with a different focus on human action: *metaethics*, *normative ethics*, *applied ethics*, and *descriptive ethics* (Stahl, 2013). We provide an overview on these disciplines, subject areas, and methodological foundations in Table 1.

[2]As also reflected by the many top-tier NLP conferences with dedicated ethics submission tracks, e.g., *EMNLP 2023*.

**Morality.** Examining ethical frameworks brings us to the concept of *morality* itself, which is defined differently depending on the ethical perspective at hand. The concrete definition is crucial for ethical reflection in language technology, as 'morality' can be used in both a descriptive and a normative sense. In the normative sense, morality is seen as a set of principles that govern human behaviour (Strawson, 1961), or as a socially constructed concept shaped by cultural and individual perspectives (Gert and Gert, 2020). In the descriptive sense, however, 'morality' refers "to certain codes of conduct put forward by a society or a group (such as a religion), or accepted by an individual for her own behaviour" (Gert and Gert, 2020).

**Metaethics.** We refer to the ethical branch which provides the analytical foundations for the other three sub-disciplines (*normative*, *applied*, and *descriptive ethics*) as metaethics. It is concerned with the universal linguistic meanings of the structures of moral speech as well as the power of ethical theories (Sayre-McCord, 2023), and deals with general problems underlying ethical reasoning, like questions around moral relativism.

**Normative Ethics.** This sub-discipline investigates universal norms and values as well as their justification (Copp, 2007). We operate within the normative framework if we make moral judgements and evaluate an action as right or wrong. It thus represents the core of general ethics and is often referred to as *moral philosophy* or simply *ethics*.

**Ethical Theories and their Families.** Within normative ethics, philosophers have presented various reasoning frameworks, dubbed *ethical theories*, that determine whether and why actions are right and wrong, starting from specific assumptions (Driver, 2022). These theories can be – in western philosophy – roughly assigned to three competing ethical families (or are hybrids): virtue ethics, deontological ethics, and consequentialism. While *virtue ethics* focuses on cultivating the moral character and integrity of the person guiding moral action (Hursthouse and Pettigrove, 2022), *deontological ethics* and *consequentialism* emphasise the status of the action, disposition or rule. Concretely, the former focuses on duty, rules and obligations, regardless of an action's consequences (Alexander and Moore, 2021), while the latter focuses on the consequences of actions and places moral value based on the outcomes (Sinnott-Armstrong, 2022).

| Discipline | Subject Area | Method |
|---|---|---|
| METAETHICS | Language and logic of moral discourses, moral argumentation methods, ethical theories' power | Analytical |
| NORMATIVE E. | Principles and criteria of morality, criterion of morally correct action, principles of a good life for all | Prescriptive, abstract judgement |
| APPLIED E. | Valid norms, values, and recommendations for action in the respective field | Prescriptive, concrete judgement |
| DESCRIPTIVE E. | Followed preferences for action, empirically measurable systems of norms/ values | Descriptive |

Table 1: Overview of the four different branches of ethics we describe (metaethics, normative ethics, applied ethics, descriptive ethics). We characterise the subject area and name the underlying methodological root.

**Applied Ethics.** Applied ethics builds upon the general normative ethics framework but deals with individual ethics in concrete situations (Petersen and Ryberg, 2010). This includes, for example, *bioethics*, *machine ethics*, *medical ethics*, *robot ethics*, and the *ethics of AI*.

**Descriptive Ethics.** The aforementioned subbranches of ethics starkly contrast with descriptive ethics (which is why it is not always counted among the main disciplines of ethics). Descriptive ethics represents an empirical investigation and describes preferences for action or empirically found systems of values and norms (Britannica, 2023). The most important distinction from the previous two disciplines is that it does not make moral judgements and merely describes or explains found criteria within a society, e.g., via surveys such as the World Value Survey (Haerpfer et al., 2020).

**Moral Psychology.** Finally, we distinguish between moral philosophy and *moral psychology*. As mentioned, moral philosophy can be understood as normative ethics and thus deals with the question of right action and represents a judgemental action. In contrast, moral psychology relates to descriptive ethics. It explores moral judgements and existing systems of values and norms to understand how people make moral judgements and decisions. This distinction is crucial, as many models and methods covered in our survey refer to the *Moral Foundation Theory* (MFT). This social psychology theory aims to explain the origin and variation of human moral reasoning based on innate, modular foundations (Graham et al., 2013, 2018).

## 3 Survey Methodology

Our approach to surveying works dealing with morality in NLP consists of three steps: (1) scope definition and paper identification, (2) manual analysis of the relevant papers, and (3) validation.

### 3.1 Search Scope and Paper Identification

To identify relevant publications, we queried the ACL Anthology,[3] Google Scholar,[4] and the ACM Digital Library[5] for the following keywords: *'consequentialism', 'deontology', 'deontological', 'ethical', 'ethics', 'ethical judgement', 'moral', 'moral choice', 'moral judgement(s)', 'moral norm(s)', 'moral value(s)', 'morality', 'utilitarianism', 'virtues'.*[6] We conducted this initial search between 25/01/2023 and 27/01/2023. For each engine, we considered the first 100 search results (sorted by relevance) and made an initial relevance judgement based on the abstract. After removing duplicates, we ended up with 155 papers. Since our survey is limited to papers that deal with morality in the context of NLP, we examined these 155 papers more closely concerning our topic of interest (e.g., by scanning the manuscript's introduction and checking for *ethics buzzwords*) during multiple rounds of annotation. Of the original 155 papers, we identified 71 as irrelevant. We have, for example, classified as "irrelevant" papers that deal with judicial judgements, ethical decisions in autonomous vehicles, meta-analyses in NLP, and papers that deal with ethical issues in NLP on a general level or that have no particular relation to NLP. This left us with 84 remaining publications fitting our scope. Based on the references provided in this initial set, we expanded our set by eight more papers, leading us to a list of 92 papers.

### 3.2 Manual Analysis

Next, we analysed our collection manually. To this end, we developed an annotation scheme consisting of four main aspects, which we iteratively

---

[3] https://aclanthology.org
[4] https://scholar.google.com
[5] https://dl.acm.org
[6] For Google Scholar and ACM Digital Library searches, we added the keyword 'NLP'. E.g., instead of 'consequentialism', we searched for 'consequentialism NLP' to narrow down the retrieved papers to those fitting our search scope.

refined during the analysis (e.g., adding a subcategory whenever we found it necessary):

*Goal*: What is the overall goal of this work? Do authors tackle a specific NLP task?

*Foundation*: Do authors mention a theoretical foundation as basis for their work? If yes, which and to which family of thought does it belong to (e.g., moral psychology vs. moral philosophy)?

*Terminology*: Do authors use terms stemming from philosophy? How? Do they provide definitions?

*Data*: What data is used? What is the origin of this data, and which languages are represented?

We provide the full scheme with all sub-codes in the Appendix D. We conducted the analysis in multiple stages, from more coarse-grained to more fine-grained, re-analysing the papers when adding a new label. We relied on the qualitative data analysis software *MAXQDA Analytics Pro 2022* to support the whole process. After four rounds of analysis, we ended up with 4,988 annotations.

### 3.3 Validation

To ensure the validity of our results, we designed an additional annotation task, for which we devised a series of 14 questions dealing with the most important aspects of our analysis. For instance, we ask whether a publication is *relevant* for our analysis, whether it discusses the underlying *philosophical* or *psychological foundations*, whether it proposes a new framework to *measure morality*, analyses *moral sentiment*, etc. We explain the whole design of this task in Appendix B. We hired two annotators for the task who are proficient English speakers and explained the terminology we adhere to and the annotation task to them. Next, we randomly sampled 25 papers from our collection and assigned ten and fifteen respectively to each annotator. We compared the annotators' answers to our analysis and obtained an inter-annotator agreement of 0.707 Krippendorff's $\alpha$ (Krippendorff, 2011) (computed over 229 annotations) indicating *substantial agreement* (Landis and Koch, 1977).

### 4 The Status Quo

We describe our findings.

**Overall Findings.** We show the 92 papers surveyed, sorted by year of publication (Figure 2) and provide a diachronic analysis of paper goals (Figure 3). Most papers were published after 2018,

| NLP Tasks | Num. Papers |
|---|---|
| Value Prediction | 24 |
| Data Set Introduction | 14 |
| Quantification | 11 |
| Ethical Advisor | 9 |
| Moral Sentiment** | 9 |
| Moral Bias* | 6 |
| Alignment of Moral Values* | 4 |
| Ethical Judgement | 4 |
| Moral Stance** | 4 |
| Analysis of Models* | 3 |
| Moral Decision Making* | 2 |
| Ethics Classification** | 1 |
| Generation of Moral Text* | 1 |

Table 2: The different tasks covered by the 92 papers. Tasks marked with '*' represent those covered under the category 'Other', while tasks with '**' are from the category 'Classification' in §4.

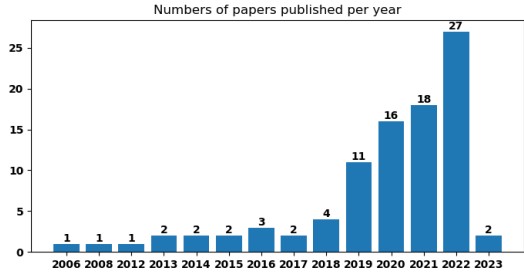

Figure 2: Distribution of the 92 surveyed papers, sorted by year of publication.

with the maximum number (27) published in 2022 – morality is a trendy topic in NLP. The first paper on morality in NLP was published in 2006, already then dealing with providing *'ethical advice'*. Overall, we observe a variety of such goals, which we classified into 13 categories (see Table 2; an extensive list of the papers falling into the different tasks can be found in Appendix A).

Out of the 92 works, more than one quarter (24) deal with *predicting moral values from text* (e.g., Pavan et al., 2023; Gloor et al., 2022) and 14 papers deal with *classification* more broadly (e.g., *'ethics classification'* (Mainali et al., 2020), *classification of ethical arguments* according to the three ethical families, and *'moral sentiment'* and *'moral stance'* classification (e.g., Mooijman et al., 2018; Garten et al., 2016; Botzer et al., 2022)), and thus fall under the umbrella of *descriptive ethics*. Another 14 papers focus primarily on the production of *'moral data sets'*, either based on MFT (e.g., Matsuo et al.,

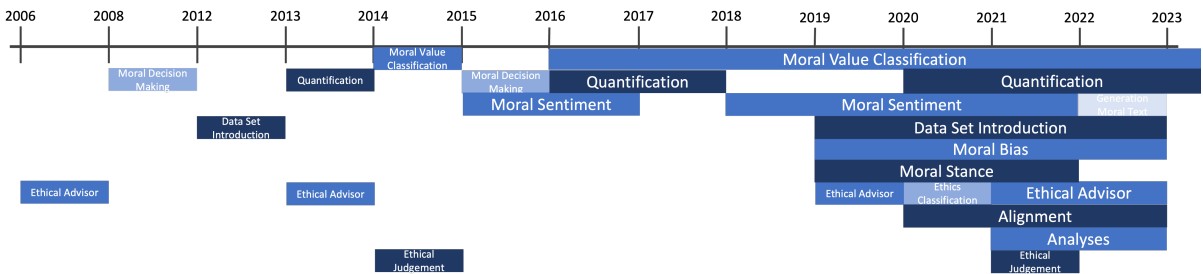

Figure 3: Timeline of which morality-related tasks were dealt with and published in NLP. Different colours are chosen for better readability and differentiation of the various categories and have no further meaning.

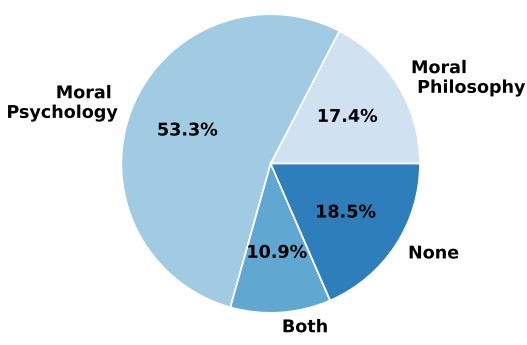

Figure 4: Foundations of studying morality. Most papers (59) mention concepts or theories from *Moral Psychology* while only 26 documents mention concepts or terms from *moral philosophy*. Ten papers use words from both domains, and 17 mention neither moral psychology nor moral philosophy.

2019; Hopp et al., 2020) or of a more general nature (e.g., Hendrycks et al., 2020; Lourie et al., 2021; Hoover et al., 2019). Twelve papers fall into *'quantification'*. This includes approaches which, e.g., based on moral psychological approaches, establish further metrics for *'measuring morality'* (e.g., *'moral concern'*, *'moral inclination'*, *'moral intensity'* (e.g., Sagi and Dehghani, 2013; Zhao et al., 2022; Kim and Lee, 2020)) or measure the *'ethicality'* of a text. In addition, nine papers present models providing moral advice judging actions as *'ethical advisors'* (e.g., Zhao et al., 2021; Jin et al., 2022; Jiang et al., 2021a) and four papers deal with models making normative judgements based on descriptive data (e.g., Yamamoto and Hagiwara, 2014; Alhassan et al., 2022).

**Foundations of Studying Morality.** We identified varying foundations pertaining to the works we surveyed (cf. Figure 4). Overall, 59 papers mention at least one moral psychology framework. Out of these, 49 base their approach on the *Moral Founda-*

*tion Theory* (MFT) (e.g., Fraser et al., 2022; Hämmerl et al., 2022b; Stranisci et al., 2021; Hoover et al., 2019; Alshomary et al., 2022; Mutlu et al., 2020), while six (also) rely on *Schwartz' Values Theory* (e.g., Kiesel et al., 2022; Gloor et al., 2022; Maheshwari et al., 2017) and one on *Kohlberg's Theory* (Rzepka and Araki, 2012). Eight documents mention moral psychology in general but do not state a specific framework. Our analysis yields 26 publications mentioning one of the ethical theories we describe above (consequentialism, deontology, virtue ethics), while just 16 go further into detail. Six documents mention aspects related to moral psychology as well as to ethical theories (Fraser et al., 2022; Alfano et al., 2018; Botzer et al., 2022; Dehghani et al., 2008; Mainali et al., 2020; Jiang et al., 2021b). In contrast, Rzepka and Araki (2015) mention that they decided to exclude ethical theories from their study and base their approach solemnly on commonsense knowledge about norms. Teernstra et al. (2016) state that MFT is an ethical framework. This is, as we outline above, not true since it is a theory of moral psychology and not moral philosophy (which provides ethical principles to what is right and wrong). To sum up, **we find that there is a lack of clarity and consistency as to whether morality in NLP is addressed purely empirically or also normatively. This lack of clarity persists also in regards to the further usage of ethical terminology**.

**Usage of Philosophical Terms.** We conduct an even finer-grained analysis of how philosophical terms are used. In total, we note that most papers (66.3%) do not define the terminology they adopt (61 papers vs. 31 papers). Some works seem to use the terms *"moral"* and *"ethics"* interchangeably (Jentzsch et al., 2019; Schramowski et al., 2020). For instance, Penikas et al. (2021) want "to assess the moral and ethical component

of a text". Similarly, we found some use *"morality"* as a synonym of *"value"* and *"moral foundation"* (Rezapour et al., 2019b,a; Lan and Paraboni, 2022; Huang et al., 2022; Liscio et al., 2022). We provide an extensive list of definitions in Table 3. NLP literature also introduces novel terms for which, sometimes, definitions are lacking. As such Hopp et al. (2020) introduce *"moral intuition"* but leave unclear what exactly they mean. (Xie et al., 2020) introduce *"moral vignette"*, possibly referring to moral values or norms, but do not provide a definition. Importantly, some authors state they base their work on applied or descriptive ethics but ultimately provide normative judgements with their models when using them to predict (or judge) new, unseen situations (Ziems et al., 2022; Forbes et al., 2020; Lourie et al., 2021; Hendrycks et al., 2020; Schramowski et al., 2022; Zhao et al., 2021; Jiang et al., 2021a,b; Botzer et al., 2022; Yamamoto and Hagiwara, 2014; Alhassan et al., 2022). This is problematic, since here, normative judgements are made from empirical data or without any normative justification. We conclude that **clear definitions of the terminology are mostly lacking**.

**Underlying Data.** In total, we identify 25 different data sets that underpin the studies we analyse (see Appendix C). Partially, however, those corpora are derivatives of each other, e.g., Matsuo et al. (2019), Hopp et al. (2020), and Araque et al. (2020) all extend the Moral Foundations Dictionary (MFD).[7] As a result, some of the data sets are very popular and widely used in the respective subfields they relate to (e.g., the original MFD is used in 35 publications). Similarly, we observe heavy reliance on social media data: Twitter is used in 32 publications (e.g., Hoover et al., 2019; Stranisci et al., 2021) and in nine papers, researchers rely on Reddit data (e.g., Trager et al., 2022; Alhassan et al., 2022). As previously observed in NLP (Joshi et al., 2020), the distribution of languages the works we survey deal with are highly skewed (see Figure 5). Out of 33 papers that explicitly state the language they deal with, we find that the vast majority (75.8%) deal with English. Also, the interrelation of multilinguality and morality is still underresearched with only four papers dealing with more than one language (Hämmerl et al., 2022a,b; Guan et al., 2022; Lan and Paraboni, 2022). To conclude, we find that **the data sets used are heavily skewed w.r.t. source and linguistic diversity.**

---

[7] https://moralfoundations.org

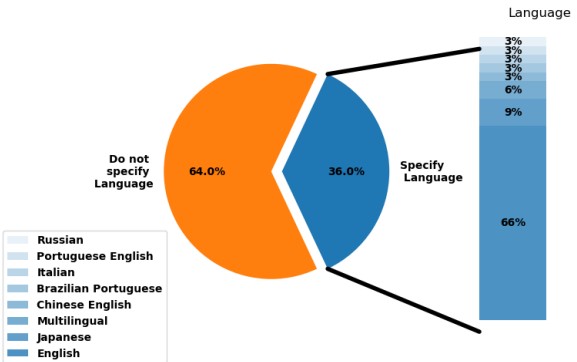

Figure 5: Distribution of languages within data sets. Only 33 publications explicitly name the language(s) used. Of these, 75.8% use English as their language.

## 5 Challenges

Based on our findings, we discuss the scientific methodological problems and the resulting ethical challenges in the current landscape of work on language technology and morality.

**Missing Foundations.** Our findings indicate that the underlying foundations of morality in NLP, as well as the respective terminologies, are diverse but often unclear and left implicit. The foundations are, however, a crucial aspect of these studies: there exist different definitions of morals (and values, norms, etc.) and what they imply. Our findings show that the underlying foundations of morality in NLP and the corresponding terminologies are diverse but often unclear and remain implicit. However, the foundations are a crucial aspect of these studies: there are different definitions of morality (and values, norms, etc.) and what it implies. Consequently, different disciplines may have a completely different focus, such as in the distinction of moral psychology vs. moral philosophy, where the descriptive and normative bases are contrasted. This distinction is crucial because of the following implications, as already outlined in §2. Within moral philosophy, we must continue to compare different ethical theories as they may compete with each other (e.g., deontology vs. consequentialism). We can draw parallels to the field of *Affective Computing*[8] here: theories on emotions are similarly diverse (e.g., James, 1948; Darwin, 1999; Mellers et al., 1997; Scherer et al., 2001) and similarly influence the research outcome (cf. Barrett et al., 2019; Mau et al., 2021). However, we find that, in affec-

---

[8] Rosalind Picard defines Affective Computing as 'computing that relates to, arises from, or deliberately influences emotions' (Picard, 2000, p. 3).

| Paper | Foundation | Definition | Concept |
|---|---|---|---|
| (Schramowski et al., 2022) | | "**morality** has referred to the "right" and "wrong" of actions at the individual's level, i.e., an agent's first-personal practical reasoning about what they ought to do." | right and wrong of actions |
| (Roy et al., 2021) | MFT | **Morality** is "a set of principles to distinguish be-tween right and wrong" | set of principles |
| (Jiang et al., 2021a) | MFT, P | "Philosophers broadly consider **morality** in two ways: morality is a set of objectively true principles that can exist a priori without empirical grounding (Kant, 1785/2002; Parfit, 2011); and morality is an expression of the biological and social needs of humans, driven by specific contexts (e.g., time and culture, Smith, 1759/2022; Wong, 2006; Street, 2012)." | set of principles and expression of needs |
| (Jiang et al., 2021b) | MFT, P | "formalize **morality** as socially constructed expectations about acceptability and preference." | expectations |
| (Lan and Paraboni, 2022) | MFT | **morality** is "a system of values and principles that determines what is admissible or not within a social group" | system of values |
| (Rezapour et al., 2019b) | MFT | "To extract **human values** (in this paper, **morality**) and measure social effects (morality and stance) ..." | Morality = Moral Value |
| (Rezapour et al., 2019a) | MFT | "To capture **morality** in tweets, we found and counted all words that matched entries in the enhanced MFD" | Morality = Moral Value |
| (Huang et al., 2022) | MFT | "we focus on the **morality** classification task" | Morality = Moral Value |
| (Liscio et al., 2022) | MFT | "**Morality** helps humans discern right from wrong. Pluralist moral philosophers argue that human morality can be represented, understood, and explained by a finite number of irreducible basic elements, referred to as moral values (Graham et al., 2013)." | Morality = Moral Value |
| (Asprino et al., 2022) | MFT | "**Morality** [is] a set of social and acceptable behavioral norms", "**Moral values** [are] commonsense norms shape [that] our everyday individual and community behavior." | norms |
| (Araque et al., 2020) | MFT | "**Moral values** are considered to be a higher level construct with respect to personality traits, determining how and when dispositions and attitudes relate with our life stories and narratives [27]." | dispositions |
| (Vecerdea, 2021) | MFT | **moral values** are "abstract ideas that ground our judgement towards what is right or wrong" | abstract ideas |
| (Constantinescu, 2021) | MFT | "**personal values** are the abstract motivations that drive our opinions and actions" | abstract motivations |
| (Dondera, 2021) | MFT | "**Moral values** are the abstract motivations that drive our opinions and actions." | abstract motivations |
| (Arsene, 2021) | MFT | "**Moral values** represent the underlying motivation behin[d] people's opinions, which influence their day-to-day actions." | underlying motivations |
| (Lin et al., 2018) | MFT | "**Moral values** are principles that define right and wrong for a given individual. They influence decision making, social judgements, motivation, and behaviour and are thought of as the glue that binds society together (Haidt)" | principles |

Table 3: The different definitions for *'morality'* and *'moral values'* in the papers. In the 'Foundation' column, we distinguish between Moral Foundation Theory (MFT), and Philosophy (P).

tive computing, these theories are considered and adapted, and accordingly, corresponding models are developed (Marsella et al., 2010). Thus, these studies mostly have an explicit root in certain theories of emotion psychology. In contrast, such a systematic approach is currently missing for tasks regarding morality in NLP.

**Missing Context.** Essential aspects and dimensions of morality are lost when trying to derive moral values or ethical attitudes from text alone and from incomplete textual descriptions. Moral judgements are always context-dependent, and without an accurate description of the context, valuable information is lost (Schein, 2020). Most approaches, however, disregard the broader context completely. They focus only on the presence of certain words, which, for example, are tied to specific moral values (Jentzsch et al., 2019; Lourie et al., 2021; Yamamoto and Hagiwara, 2014; Kaur and Sasahara, 2016, e.g.,). Some also focus on so-called *atomic actions*, which severely limits the ability to make an accurate judgement (Schramowski et al., 2019, 2020, 2022). This problem also relates to the data sets used. For instance, the context available in Twitter data is directly constrained by the character limit of tweets. While context-dependency and missing knowledge is a general problem in NLP (cf. Lauscher et al., 2022b), the problem is likely more severe when it comes to morality: moral models

trained on such limited data sets may introduce or reinforce existing social biases in individuals when deriving moral judgements, leading to unfair evaluations and misrepresentations of people and situations. This could have detrimental consequences for their personal and professional lives, as the beliefs about the morality of users of such moral models may be influenced.

**Missing Diversity.** Another challenge is that there is no universal ground truth for moral judgements and ethics in general (yet). Morality is the subject of constant philosophical and cultural debate and has evolved. Although Aristotle defined the concept of ethics already ca. 350 B.C.E. (in the Western philosophical tradition) as the branch of philosophy concerned with habits, customs and traditions (Aristotle, ca. 350 B.C.E/2020), to this day, there is no universally accepted ethical framework that defines *the one ethic* as the 'right' one (Aristotle, ca. 350 B.C.E/2020). Consequently, subjective interpretations of moral concepts, often used as a basis for training data, can vary depending on the individual, cultural and societal circumstances (Alsaad, 2021; Driver, 2022). This recognition stands in stark contrast to the heavily skewed data sets available. For instance, as we showed, languages other than English have been mostly ignored, and data sets are mostly based on two(!) social media platforms, which, making things worse, mostly at-

tract male users from the USA (Ruiz Soler, 2017; Proferes et al., 2021). This suggest a severe lack of cultural and sub-cultural diversity.

**Is-ought Problem.** Research on morality in NLP often aims at extracting normative judgements from their empirical analyses (Jiang et al., 2021a,b; Shen et al., 2022; Yamamoto and Hagiwara, 2014; Efstathiadis et al., 2022; Alhassan et al., 2022; Schramowski et al., 2022; Lourie et al., 2021; Forbes et al., 2020; Ziems et al., 2022). In doing this, they face the so called *is-ought problem* (Cohon, 2018): it is not ethically legitimate to derive normative judgements from empirical observations – is does not imply ought. Put differently: just because many people think something is morally right, does not mean it is ethically justified. Normative judgements require grounding in ethical theories or principles that go beyond the mere observation of language use (Cohon, 2018). Without a clear ethical theory to guide the derivation of normative judgements, models may inadvertently perpetuate biases or reinforce existing social norms, leading to unjust or discriminatory outcomes. Especially when subjective judgements are used as the basis instead of ethical theories, specific biases may be unintentionally imposed by relying exclusively on the patterns or norms in the data. Such an approach does not consider legitimate differences in moral reasoning and results in a narrow and biased understanding of normative judgements.

Overall, we conclude that including morality in NLP models, not only limited to making moral judgements, is a constant challenge. The 'is-ought' problem, the lack of ethical foundations, contextual complexity, subjectivity and pluralism highlight our current limitations and potential pitfalls.

# 6 Recommendations

We propose three recommendations (**R's**) to help research with avoiding the pitfalls described above.

*(R1) Integrate fundamental ethics approaches into NLP systems that explore ethical theories.* Moral philosophical approaches provide well-established foundations for understanding and evaluating ethical principles and judgements. Only by incorporating established foundations and theories we can develop a more robust framework that goes beyond a purely descriptive analysis. This will allow for a more comprehensive and nuanced exploration of moral issues and facilitate the development of language models consistent with widely accepted ethical theories. At the same time, it will also maintain the ethical consistency of language models in decision-making processes. Using philosophical foundations ensures that the moral judgements automatically made are consistent with consistent principles and avoid contradictory or arbitrary assessments. Furthermore, as ethical theories often emphasise the importance of context and recognise the diversity of moral values and perspectives, we will promote the analysis of moral judgements in context and avoid over-generalisations or biased interpretations.

*(R2) Include and explicitly name ethical theories to which the model refers, as well as terms that come from philosophy when used otherwise.* The explicit use and naming of underlying ethical theories creates clarity and ensures consistency in moral discussions in NLP. By naming specific approaches, researchers and users can create a common language and framework for morality in NLP. This promotes a shared understanding of the underlying principles and concepts, enabling more effective communication and collaboration. Incorporating ethical theories into language technology research also allows researchers to conduct more robust analyses of moral judgements, considering different perspectives and applying established criteria for ethical evaluation. It also prompts ethical reflection and examination. By explicitly naming ethical theories, researchers are encouraged to reflect on the extent to which their research or (computational) model conforms to or deviates from these theories, further promoting ethical awareness and accountability. Importantly, the explicit use of ethical theories and a shared terminology will facilitate interdisciplinary collaboration between NLP researchers and ethicists. By using established ethical theories and definitions of the relevant terminology, researchers from different disciplines can effectively communicate with each other, bridge gaps, and draw on expertise from multiple fields. This collaboration can thus lead to more comprehensive and informed research findings.

*(R3) Use a consistent vocabulary regarding crucial terms such as 'ethics', 'morality', 'values' or 'norms'. Define introduced terms and check whether the terminology has been used in the literature before.* Consistent vocabulary brings clarity and precision to discussions and research on morality in NLP. Researchers can thus effectively communicate their ideas, findings, and arguments

using well-defined and commonly accepted terms. This helps avoid confusion or misinterpretation between scholars and readers and facilitates accurate knowledge exchange. A uniform terminology also ensures conceptual alignment with the existing literature. Established terms allow researchers to build on previous research and link their work to a broader, more interdisciplinary body of knowledge.

## 7    Related Work

There exist a plethora of works dealing with ethical issues and the social impact of NLP (e.g., Hovy and Spruit, 2016; Leidner and Plachouras, 2017; Parra Escartín et al., 2017; Lauscher et al., 2022a; Hessenthaler et al., 2022; Kamocki and Witt, 2022, *inter alia*). Accordingly, in this realm, researchers also have provided systematic overviews of the literature, e.g., on 'bias' in NLP (Blodgett et al., 2020) and 'ethics' within the NLP community (Fort and Couillault, 2016). North and Vogel (2019) presented a categorisation of ethical frameworks of NLP into different ethical families. Yu et al. (2018) took a closer look at technical approaches for ethical AI and provide a taxonomy for the field of AI Governance. Closest to us, Hagendorff and Danks (2022) presented a meta-view of moral decision-making in AI outlining ethical and methodological challenges, focusing, like Talat et al. (2022) on the example of DELHPI (Jiang et al., 2021a).

## 8    Conclusion

In reviewing 92 papers dealing with morality in NLP, we found that (a) the majority of the papers do not use ethical theories as a basis but predominantly take descriptive approaches, whereby judgements derived from them are subject to the 'is-ought' problem; (b) relevant terms such as 'moral', 'ethics' or 'value' are often not properly defined nor distinguished; and (c) explanatory definitions are rarely provided. Based on our analysis, we then provided three recommendations to help researchers avoid the resulting pitfalls. These recommendations involve a stronger integration of philosophical considerations to guide the field in a more targeted and sound direction.

## Acknowledgements

This work is in part funded under the Excellence Strategy of the Federal Government and the Länder. We thank the anonymous reviewers for their insightful comments.

## Limitations

We recognise that this work is limited in several aspects. First, the papers we consider are determined using the selected databases and the English language. Furthermore, our foundational philosophical chapters are based on a Western understanding, which means that our definitions were developed within Western academic traditions and therefore have the limitations that come with them. Through the papers analysed, we have also intensely focused on the widely cited "Moral Foundation Theory" of Graham and Haidt, which is why other theories of moral psychology have been neglected. Future papers may therefore address and analyse other moral psychological and moral philosophical theories in NLP. As part of our analysis, we have only limited ourselves to the are of NLP, which the selection of our databases and papers already shows. Accordingly, the results presented in this paper relate only to the are of NLP and not other AI/ML related fields. Finally, it should be noted that our recommendations are not comprehensive and should be used to develop further questions and strategies.

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

# A Categorisation details

## A.1 Alignment

Papers in the category 'alignment' deal with the moral orientation of AI. Human moral values are taken as the basis on which the alignment should take place. (Ammanabrolu et al., 2022; Hendrycks et al., 2021; Liu et al., 2022; Tay et al., 2020)

## A.2 Analyses

This work primarily relates to the 'Delphi Model' (Jiang et al., 2021a,b) and analyses the approach to making automatic moral judgements. (Fraser et al., 2022; Talat et al., 2021, 2022)

## A.3 Data Sets

Within this category are papers that focus primarily on constructing a 'moral data set'. These papers can be divided into two approaches. On the one hand, papers based on Moral Foundation Theory (Graham et al., 2013) extend the Moral Foundation Dictionary. (Matsuo et al., 2019; Hopp et al., 2020; Araque et al., 2020)

A second group of papers deals with more general datasets, such as Twitter, Reddit, or datasets created by annotators. All documents in this superordinate category have in common that they are in some way related to moral values, norms or ethics and can be used for morality in NLP. (Guan et al., 2022; Hendrycks et al., 2020; Stranisci et al., 2021; Rzepka and Araki, 2012; Sap et al., 2020; Hoover et al., 2019; Emelin et al., 2021; Lourie et al., 2021; Forbes et al., 2020; Trager et al., 2022; Ziems et al., 2022)

## A.4 Ethical Advisor

Papers in this category present models intended to act as ethical advisors. These papers have in common that they propose a model that should be able to make moral decisions and advise the user on whether the decision is 'good' or 'bad'. The models in this category are primarily trained on descriptive approaches and are then supposed to be able to make normative judgements. (Zhao et al., 2021; Peng et al., 2019; Bang et al., 2022; Jiang et al., 2021a,b; Gu et al., 2022; Komuda et al., 2013; Jin et al., 2022; Anderson et al., 2006)

## A.5 Ethical Judgement

Papers in this category present models trained to make moral judgements based on descriptive data and produce their normative judgements as output.

Similar to papers in the 'Ethical Advisor' category, this category also faces the 'is-ought' problem. (Shen et al., 2022; Yamamoto and Hagiwara, 2014; Efstathiadis et al., 2022; Alhassan et al., 2022)

## A.6 Ethics Classification

Papers in this category are concerned with classifying moral reasoning within the three prominent families of ethics, deontology, consequentialism and virtue ethics. (Mainali et al., 2020)

## A.7 Generation of Moral Text

This category includes papers that deal with generating and analysing moral arguments. (Alshomary et al., 2022)

## A.8 Moral Bias

Papers in this category represent work to analyse and map the moral bias of large language models such as BERT (Devlin et al., 2018). Due to the underlying training data, language models have their own 'moral compass', which can be mapped. At the same time, the approaches are to be used to reduce moral bias. (Schramowski et al., 2019; Jentzsch et al., 2019; Hämmerl et al., 2022a; Schramowski et al., 2022; Hämmerl et al., 2022b; Schramowski et al., 2020)

## A.9 Moral Decision Making

Papers in the category 'Moral Decision Making' are primarily concerned with modelling the process of moral decision-making and attempting to reconstruct it. These papers propose frameworks on how to model moral decisions. (Hromada, 2015; Dehghani et al., 2008)

## A.10 Moral Sentiment

These papers analyse moral sentiment and thus focus on the emotional polarity of a text or statement. This usually involves a classification into 'positive', 'neutral', and 'bad'. (Rzepka and Araki, 2015; Mooijman et al., 2018; Ramezani et al., 2021; Garten et al., 2016; Otani and Hovy, 2019; Xie et al., 2019; Qian et al., 2021; Kobbe et al., 2020; Roy et al., 2021)

## A.11 Moral Stance

Papers dealing with moral stances focus on expressing the speaker's point of view and judgment concerning a particular statement. These papers focus on identifying a person's moral standpoint on a topic. (Roy and Goldwasser, 2021; Botzer

et al., 2022; Santos and Paraboni, 2019; Pavan et al., 2020)

## A.12 Quantification

Under the broad category of 'quantification' fall all papers that measure 'morality' or 'ethics' in some way. These papers cannot be assigned to one of the other categories, as new terms and metrics are often introduced. The only thing they have in common is the quantification of 'morality'. (Kim and Lee, 2020; Zhao et al., 2022; Sagi and Dehghani, 2013; Mutlu et al., 2020; Hulpuș et al., 2020; Nokhiz and Li, 2017; Penikas et al., 2021; Kennedy et al., 2021; Xie et al., 2020; Xu and Guo, 2023; Kaur and Sasahara, 2016)

## A.13 Value Prediction

All work to classify moral values falls under this category. Most of these papers are based on Moral Foundation Theory, with a few exceptions (see § 4). (van den Broek-Altenburg et al., 2021; Altuntas et al., 2022; Rezapour et al., 2019b; van Luenen, 2020; Rezapour et al., 2019a; Vecerdea, 2021; Lan and Paraboni, 2022; Asprino et al., 2022; Constantinescu, 2021; Arsene, 2021; Dondera, 2021; Lin et al., 2018; Huang et al., 2022; Liscio et al., 2022; Pavan et al., 2023; Mokhberian et al., 2020; Teernstra et al., 2016; Johnson and Goldwasser, 2019; Maheshwari et al., 2017; Gloor et al., 2022; Alfano et al., 2018; Dahlmeier, 2014; Kiesel et al., 2022; Johnson and Goldwasser, 2018)

## B  Validation Task

1. Is the topic of the paper related to the concepts of 'natural language processing' (NLP) and 'morality'? (e.g. the paper uses methods or algorithms of NLP and deals with for example moral judgement, values, norms, morality, or ethics)
   - yes no

2. Does the paper state/use any philosophical foundations (e.g. underlying ethics family <deontology, consequentialism, virtue ethics>, definitions of morality or familiar terms used)?
   - yes (please specify) no

3. Does the paper state/use any psychological foundations (e.g. 'Moral Foundation Theory' or 'Schwartz Value Theory')
   - yes (please specify) no

4. Does this paper deal with the **classification** of moral values, norms or other concepts related to 'morality' in general?
   - yes (please specify what is classified) no

5. Does the paper propose a new **framework to measure or quantify morality or related concepts**?
   - yes (please specify the name of the proposed framework and what is quantified) no

6. Does the paper investigate **ethical or moral bias** in models?
   - yes no

7. Is the paper concerned with the **alignment of human values**? (e.g. does the paper use morality or moral values as a way to align AI with human values?)
   - yes no

8. Does the paper analyse **moral sentiment** or **moral stance**?
   - yes no

9. Does the paper try to **model moral decision making**?
   - yes no

10. Does this paper present some kind of **ethical advisor**, i.e. an algorithm which is able to answer questions relating to morality or generate moral judgements?
    - yes no

11. Does the paper make any **predictions regarding human values or moral judgement** which go beyond mere classification of such? (E.g. is the model able to make its own moral judgements?)
    - yes (please specify in what ways) no

12. Does the paper introduce a new **data set**?
    - yes (please name) no

13. Which data source(s) does the paper use?
    - Twitter Reddit MFD other (please specify) not stated

14. Which language(s) are processed?
    - English other (please specify)

## C Overview of Datasets used

| Dataset | Used in |
| --- | --- |
| ANECDOTES (Lourie et al., 2021) | (Shen et al., 2022) |
| BR Moral Corpus (Pavan et al., 2020) | (Lan and Paraboni, 2022) |
| DILEMMAS (Lourie et al., 2021) | (Shen et al., 2022) |
| ETHICS (Hendrycks et al., 2020) | (Jiang et al., 2021a,b; Liu et al., 2022; Gu et al., 2022) (Ammanabrolu et al., 2022) |
| extended Moral Foundation Dictionary (Hopp et al., 2020) | (Mutlu et al., 2020; Ziems et al., 2022; Rezapour et al., 2019a) |
| Helpful, Honest & Harmless (Askell et al., 2021) | (Liu et al., 2022) |
| Japanese Lexicon | (Rzepka and Araki, 2012) |
| Japanese MFD | (Matsuo et al., 2019) |
| MACS | (Tay et al., 2020) |
| MoralConvIta | (Stranisci et al., 2021) |
| MoralExceptQA | (Jin et al., 2022) |
| Moral Foundation Dictionary https://moralfoundations.org/ | (Mutlu et al., 2020; Xie et al., 2019; Qian et al., 2021; Hulpuș et al., 2020) (Kennedy et al., 2021; van den Broek-Altenburg et al., 2021; Nokhiz and Li, 2017) (Rezapour et al., 2019b; Lin et al., 2018) (Johnson and Goldwasser, 2018) (Rezapour et al., 2019a; Sagi and Dehghani, 2013; Penikas et al., 2021) |
| Moral Foundation Reddit Corpus | (Trager et al., 2022) |
| Moral Foundation Twitter Corpus (Hoover et al., 2019) | (Trager et al., 2022; Constantinescu, 2021; Lan and Paraboni, 2022) (Dondera, 2021; Araque et al., 2020; Asprino et al., 2022) (Ramezani et al., 2021; Huang et al., 2022) (Roy and Goldwasser, 2021; Vecerdea, 2021) (Liscio et al., 2022; van Luenen, 2020; Arsene, 2021) |
| Moral Strength | (Araque et al., 2020) |
| Moral Stories (Emelin et al., 2021) | (Gu et al., 2022; Jiang et al., 2021a,b; Liu et al., 2022) (Bang et al., 2022; Zhao et al., 2022; Ammanabrolu et al., 2022) |
| RealToxicityPrompts (Gehman et al., 2020) | (Liu et al., 2022; Schramowski et al., 2022) |
| SCRUPLES (Lourie et al., 2021) | (Jiang et al., 2021a,b; Ammanabrolu et al., 2022) |
| SOCIAL-CHEM-101 (Forbes et al., 2020) | (Gu et al., 2022; Emelin et al., 2021; Jiang et al., 2021a,b) (Ziems et al., 2022; **?**; Bang et al., 2022; Shen et al., 2022; Ammanabrolu et al., 2022) |
| SOCIAL BIAS INFERENCE CORPUS (Sap et al., 2020) | (Jiang et al., 2021a,b; Ammanabrolu et al., 2022) |
| STORAL | (Guan et al., 2022) |
| Story Commonsense (Rashkin et al., 2018) | (Gu et al., 2022) |
| TrustfulQA (Lin et al., 2021) | (Liu et al., 2022) |

Table 4: Overview of the different datasets used.

# D  Overview of Labels

| Label | Sub-Labels | N |
|---|---|---|
| Data Sets | ANNECDOTES, AITA Dataset, BR Moral Corpus, Common Sense Norm Bank, DILEMMAS, ETHICS, Helpful, Honest & Harmless, MACS, MFD, Moral Integrity Corpus, Moral Stories, MoralConvITA, MoralExceptQA, NYT, RealToxicityPrompts, ROCStories, SCRUPLES, Social Bias Inference Corpus, Social-Chem-101, STORAL, Story Commonsense, Trustful QA | 214 |
| Definitions | Definitions, No Definitions | 107 |
| Ethics Family | Consequentialism, Deontology, Virtue Ethics, ANY: Virtue Ethics Deontology Consequentialism (AC) | 228 |
| Goals | Classification, Data Set Introduction, Dimensions, Generation, Framework, Prediction | 550 |
| Interesting Passages | | 176 |
| Keywords | Consequentialism, Deontology, Ethical/Ethics, Ethical Judgement, Inductive Paper, Moral Choice, Moral Judgement, Moral NLP, Moral Norms, Moral NLP, Morality, Utilitarianism, Virtues | 285 |
| Language | | 38 |
| Methodology | Assumptions, Caution Statement, Motivation | 250 |
| Model | | 106 |
| Moral Psychology | Cheng & Fleischmann, DOSPERT Values, Kohlberg's Theory, Moral Foundation Theory, NEO FFI-R personality survey, Schwartz Values Theory, Shweder Big Three, Moral Foundation Theory (AC) | 231 |
| Philosophical Terms | Applied Ethics, Common Sense Knowledge, Descriptive Ethics, Ethical X, Ethics, Moral, Moral X, Morality, Moral Philosophy, Normative Ethics, Norms, Virtue/Vice | 748 |
| Proposed Frameworks | AISocrates, CAMILLA, Delphi, DREAM, Frame-based Value Detector Model, GALAD, Jimminy Cricket Environment, Moral Choice Machine, MoralCOT, MoralDirection Framework, MoralDM, MoralScore, Morality Frames, Ned, Neural Norm Transformer, Project Debater, SENSEI, The Morality Machine | 46 |
| Results | | 30 |
| Sources | Applied Ethics Literature, ArgQuality Corpus, Blogs/Open Web, Dear Abby Advice, E-Mails, Facebook, Kaggle, Newspaper, Quarr, Reddit, Yahoo Japan, Yelp, Reddit (AC), Twitter (AC) | 2,047 |
| | | 4,988 |

Table 5: Overview of our used annotation scheme. Sub-Labels with an '(AC)' indicate labels generated by the auto-code function of *MAXQDA2022*.