# OpenReview forum: "Values, Ethics, Morals? On the Use of Moral Concepts in NLP Research"
_EMNLP/2023/Conference — EMNLP 2023 Findings_

### Official Review · Reviewer_fP8o · 2023-08-03

**Soundness:** 3

**Excitement:**

2: Mediocre: This paper makes marginal contributions (vs non-contemporaneous work), so I would rather not see it in the conference.

**Paper Topic And Main Contributions:**

The work features a statistical survey based upon mentions of moral concepts in 92 research papers published from 2006 to 2023. The work also proposes a summary of those concepts, by drawing upon ancient and contemporary philosophy.

The survey uses data analysis software as well as the help of two human annotators, who were asked (mostly yes-no, aka "binary") questions such as "Does the paper investigate ethical and moral bias in models?" (cf. the appendix B).

**Reasons To Accept:**

The survey classifies the 92 papers into 13 categories, such as "Data Sets" and "Moral Decision Making" (cf. the appendix A).

The survey also reports the following: more mentions of moral psychology concepts (i.e. of the Moral Foundation Theory), than mentions of moral philosophy concepts; heavy use of social media; a bias towards processing English (but not other languages); flaws in defining terminology.

**Reasons To Reject:**

I am not convinced that the proposed survey does not itself have the "missing context" problem; the survey observes this problem in others' research (cf. 5 Challenges). I mean that the survey itself apparently focuses "only on the presence of certain words" but not on the context in a broad sense, e.g. on how (morality) common sense is used in a chain of reasoning that has lead to some decision on moral values. I doubt that utilizing mainstream statistical parser software and binary questions to human annotators result in any substantial insight into deep understanding of difficult text, e.g. into understanding of moral judgment. Due to the recent abundance of works using statistical NLP (but not deep understanding NLP) to analyze pretty much anything, I would be much more excited by a deep understanding approach to moral concepts in text.

As to the survey of flaws in defining philosophical terminology, I do not find any clear explanation of what exactly "seem to use" means in "some works seem to use the terms ... interchangeably" (ll. 350-358). I mean a clear explanation of what portion of the seem-to-use conclusion comes from running the parser, what portion comes from the human annotators, and what portion comes from quoting others' research (or from some other source).

The suggestion to researchers that they should use "a consistent vocabulary" (ll. 573-588) looks raw to me. By "raw" I mean, insufficient information on how to implement the consistency in real life: checking if the terminology has been used by others before, is not enough to ensure term uniformity, because different papers use different terms.

**Reproducibility:**

N/A: Doesn't apply, since the paper does not include empirical results.

**Reviewer Confidence:**

4: Quite sure. I tried to check the important points carefully. It's unlikely, though conceivable, that I missed something that should affect my ratings.

**Typos Grammar Style And Presentation Improvements:**

The caption for Figure 2 and ll. 277-278:

A reader has to figure out that "Number of papers published per year" in the above caption and lines most likely means that out of the 92 surveyed papers, numbers of published papers are shown for each of the years (to figure that out you may as well need to add up all the publication numbers to get 92). To relieve the reader of the calculation and possible annoyance, I would include an explanation like the above or at least say "of the papers" instead of "of papers".

ll. 287-288:

"Out of the 92 works, the majority (24, more than one quarter) deals with ...": to me, "the majority" means "more than a half" and "one quarter" means something like "a considerable portion". If you agree, I suggest changing accordingly.

---

> ### Author Rebuttal · Authors · 2023-08-28
>
> We thank the reviewer for their thorough feedback!
>
> W1, W2 (use of statistical software, lack of relevant context): We thank the reviewer for their comment, which leads us to believe there is a misunderstanding. We didn't just use 'statistical parsing software' and binary questions on human annotators to arrive at our results. As described in "§3 Survey Methodology", we first did a manual, qualitative analysis of each paper. This analysis was not solely done by software alone, but only within the software. MAXQDA allows documents to be collected and organised in one place and then manually annotated with labels, comments, highlights and cross-references. As explained in '§3.2 Manual Analysis', we repeatedly read, analysed and annotated/labelled the papers. In addition, we automatically labelled frequently used terms to get a sense of the frequency of the words. This manual analysis and automatic coding finally resulted in our 4,988 individual annotation positions. In addition to this, we used two human, trained annotators to validate our previously conducted analysis. We are thus strongly convinced that our survey does not miss the relevant context.
>
> W3 (“raw” terminology suggestion): We thank the reviewer for highlighting this important aspect. We acknowledge that simply checking whether (and how) the terminology has already been used by others is insufficient to ensure consistency of terms. Instead, our proposal of a uniform vocabulary goes in a different direction. Because we, as NLP researchers, now can deal with topics such as morality, ethics or values and norms, we should also use the terminologies or definitions of the field from which we have taken these topics. On the one hand, we could create our own vocabulary for 'us' as an NLP community and redefine the terms for ourselves - but in doing so, we lose valuable opportunities to cooperate with the knowledge gained from philosophy in our work since we do not 'speak the same language'. On the other hand, we can create a (more substantial) awareness of the importance of proper terminology and take the arduous path of using language from philosophy. This will ensure that when we talk about 'morality', we are talking about the same word and not different things such as norms, values, or ethics.
>
> We thank you for your recommendation regarding the caption of Figure 2 and ll. 277 – 278 and will adjust the caption and the text accordingly.
>
> We thank all reviewers for their helpful comments, which we will incorporate in our draft revision!

---

### Official Review · Reviewer_DdBe · 2023-08-04

**Typos Grammar Style And Presentation Improvements:** 1) A legend may be helpful in Figure …
**Soundness:** 4

**Excitement:**

4: Strong: This paper deepens the understanding of some phenomenon or lowers the barriers to an existing research direction.

**Paper Topic And Main Contributions:**

The paper surveys previous NLP papers with respect to ethics and morality. Their contributions are threefold: First, the authors defined ethics concepts stemming from philosophy. Second, the authors surveyed 92 NLP papers with respect to these concepts – in particular, with respect to moral concepts. Finally the authors presented suggestions on how the NLP community should approach discussing ethical concepts in their papers. Namely, the authors recommend researchers to:

i) Integrate fundamental moral philosophies into their papers,

ii) Identify and explicitly mention these concepts in papers, and

iii) Adhere to standard meaning when using the terms such as “ethics” and “morals” to avoid confusion.

**Questions For The Authors:**

Question A: As part of your investigation, did you notice any of the issues you discovered (e.g., inconsistent use of moral concepts) arise in other AI/ML related fields? If so, how have these issues been resolved in other fields?

**Reasons To Accept:**

1) The paper provides a comprehensive overview of how NLP papers approach topics regarding morality and further ties it to fundamental philosophical concepts. This investigation (to the best of the author’s knowledge) has not been done before.

2) The paper calls for consistent use of moral concept terms in NLP literature in addition to raising issues regarding lack of diversity in current studies. This information is important to communicate to researchers working in this realm.

3) The recommendations provided in the paper are grounded by quantitative analysis.

**Reasons To Reject:**

1) The number of surveyed papers is low. However, this is likely due to increasing interest in NLP + ethics in recent years.

**Reproducibility:**

4: Could mostly reproduce the results, but there may be some variation because of sample variance or minor variations in their interpretation of the protocol or method.

**Reviewer Confidence:**

4: Quite sure. I tried to check the important points carefully. It's unlikely, though conceivable, that I missed something that should affect my ratings.

---

> ### Author Rebuttal · Authors · 2023-08-28
>
> We thank the reviewer for appreciating our work's comprehensiveness and importance!
>
> W1 (low number of papers): We thank the reviewer for their feedback. To the best of our knowledge, we surveyed all 92 relevant publications; thus, we do not believe this number to be a weakness per se. The recent observation of an increasing interest in morality is correct: As can be seen from Figures 3 and ll. 279 - 286, we have observed an increase in the number of papers. We expect that this trend will continue and that more papers will be published in the coming period with the emerging development possibilities. Therefore, it is essential to counteract the confusion of terminology.
>
> Q1: ‘As part of your investigation, did you notice any of the issues you discovered (e.g., inconsistent use of moral concepts) arise in other AI/ML related fields? If so, how have these issues been resolved in other fields?‘
> A: This is a pretty good question! Unfortunately, we don’t have an answer to it. As part of our analysis, we have only limited ourselves to the area of NLP, which the selection of our databases and papers already shows. Accordingly, our results relate only to the area of NLP. We will add this limitation.
>
> We further highly appreciate your comment on the colour coding of Figure 3 and the cut-off text. The colours in Figure 3 do not have any particular meaning and are just for better readability/extinguishing the different tasks.  We will modify the Figure accordingly so that everything is clear.

---

### Official Review · Reviewer_gSJ6 · 2023-08-05

**Soundness:** 3

**Excitement:**

3: Ambivalent: It has merits (e.g., it reports state-of-the-art results, the idea is nice), but there are key weaknesses (e.g., it describes incremental work), and it can significantly benefit from another round of revision. However, I won't object to accepting it if my co-reviewers champion it.

**Paper Topic And Main Contributions:**

The paper deals with explicating the differences between the different terms that are interchangeably used by NLP papers surrounding the topics of values, ethics, and morals. The paper follows a three-step process. First, it gives a thorough explanation of the different concepts (drawing knowledge from philosophy, psychology and others). Second, it surveys papers from NLP containing the aforementioned terms to state the current usage of the terms. Third, it draws insights from the current use to suggest potential ways to mitigate the confusion that arises from inconsistent definitions of these terms in the field of NLP.

**Questions For The Authors:**

Q1: Figure 1 does not have consistent design language and could be a bit confusing to the readers. For example, what do the dashed arrows mean?
Q2: Why is there a need to check ACM Digital Library and google scholar in addition to Anthology?

**Reasons To Accept:**

1. **Timely paper** There is a widespread confusion in NLP when it comes to topics related to values, ethics and morals. Accepting this paper is crucial in order to kickstart the conversation on resolving this confusion, by making researchers aware of the different terms and encouraging them to delve more deeply into the subjects (like philosophy) that discuss these concepts in-depth.

**Reasons To Reject:**

1. **Coverage of topics** This confusion of concepts is not limited to just values, ethics and morals, but also beliefs, norms, customs, behaviors, and ideologies. While it can be argued that these topics may deviate from values, ethics and morals, I strongly believe that they are inter-twined and often misrepresented in previous works that tie these concepts together. Drawing the line on what topics to include is difficult given the wide variety of concepts in this area, however a paper on tackling this issue should be able to have a wider coverage (or if not possible, acknowledge it in limitations with strong reasons).
2. **Framing of the paper** If the purpose of the paper is to raise awareness, the current framing of detailed survey of papers mentioning the terms is uninspiring. It is very difficult to get a sense of the bigger picture looking at the individual numbers. Ideally, the paper could first define the different terms (as is currently the case), and support their definitions with papers that use these terms correctly and incorrectly. That way, it is easier to understanding the key differences without have to read the specific numbers in prose.

**Reproducibility:**

3: Could reproduce the results with some difficulty. The settings of parameters are underspecified or subjectively determined; the training/evaluation data are not widely available.

**Reviewer Confidence:**

4: Quite sure. I tried to check the important points carefully. It's unlikely, though conceivable, that I missed something that should affect my ratings.

---

> ### Author Rebuttal · Authors · 2023-08-28
>
> We thank the reviewer for their positive feedback and appreciating the timeliness of our work!
>
> W1 (coverage of topics, e.g., “beliefs”, “norms”): We thank the reviewer for their important comment – we acknowledge the existing wider confusion about other related concepts than the ones that our submission covers. However, we strongly believe that (a) it is important to start with the most urgent issues and (b) that coverage of all possible philosophical concepts would exceed the scope of this survey. As the frequency of the terms “morals” and “values”, as well as their confusion, was highest in an initial analysis, we decided to focus on these as the most urgent topics. In case of acceptance, we will extend our manuscript by discussing this limitation.
>
> W2 (framing of the work, showcasing wrong and right usages): We thank the reviewer for this interesting suggestion! We agree that showcasing wrong and right usages of the philosophical concepts could attract more attention to our work, and we will re-frame parts of the manuscript in case of acceptance.
>
> Q1: ‘Figure 1 does not have consistent design language and could be a bit confusing to the readers. For example, what do the dashed arrows mean?‘
> A: The dashed arrows show a connection between the individual categories, which cannot be clearly distinguished. Most of the work on 'Ethical Judgement' refers to some classification of 'AITA' Reddit posts. The category 'Quantification' overlaps or connects to the classification's subcategory 'Value Prediction'. Many terms under 'moral psychology' use different terms but are strongly related to 'value prediction'. We will add this explanation to the caption.
>
> Q2: ‘Why is there a need to check ACM Digital Library and google Scholar in addition to Anthology?‘
> A: We decided to use three different sources to find as many papers for the survey as possible. Using the ACM Digital Library, we could add another eight papers to our corpus that we had not previously found in the ACL Anthology or via Google Scholar.

---

### Meta-Review · Area_Chair_2HZG · 2023-09-18

**Recommendation:** 3

**Metareview:**

This manuscript seeks to deal with the confusion that arises from the terms (and research on) values, morals, and ethics. The manuscript surveys prior literature from outside of NLP on these concepts, then surveys NLP literature, and suggests future avenues for NLP.

In general, the reviewers are positive towards this manuscript and particularly appreciate:

1. That there is confusion of the terms and this manuscript may provide a starting point to address such confusion
2. The manuscript provides a comprehensive report on how the terms have been used
3. The manuscript provides quantitive support for their recommendations and findings
4. The manuscript highlights flaws in terminology in prior work

However, the reviewers had the following contention:

1. The manuscript would do well to consider additional related terms such as norms, beliefs, customs, behaviors, and ideologies.
2. It can be hard to gain a sense of the bigger picture from how results are presented
3. The selection of papers is narrow and does not take into consideration topics where the terms in question are implicit.
4. It is unclear how to implement recommendations for common terminology in future work

Wrt. 1 both authors and reviewers agree that this manuscript can serve as a starting point for such considerations. To this effect, authors will reframe writing to make the bigger picture (2) clearer. While the authors do not address 3, the scope of addressing this is incredibly large and is in my opinion more appropriate for future work. Finally, the authors correctly identify that NLP researchers creating vocabularies may lead to further inaccuracies and limit opportunities for interdisciplinary collaboration.

---

### Decision · Program_Chairs · 2023-10-07

**Decision:**

Accept-Findings

**Comment:**

This manuscript seeks to deal with the confusion that arises from the terms (and research on) values, morals, and ethics. The manuscript surveys prior literature from outside of NLP on these concepts, then surveys NLP literature, and suggests future avenues for NLP.

In general, the reviewers are positive towards this manuscript and particularly appreciate:

1. That there is confusion of the terms and this manuscript may provide a starting point to address such confusion
2. The manuscript provides a comprehensive report on how the terms have been used
3. The manuscript provides quantitive support for their recommendations and findings
4. The manuscript highlights flaws in terminology in prior work

However, the reviewers had the following contention:

1. The manuscript would do well to consider additional related terms such as norms, beliefs, customs, behaviors, and ideologies.
2. It can be hard to gain a sense of the bigger picture from how results are presented
3. The selection of papers is narrow and does not take into consideration topics where the terms in question are implicit.
4. It is unclear how to implement recommendations for common terminology in future work

Wrt. 1 both authors and reviewers agree that this manuscript can serve as a starting point for such considerations. To this effect, authors will reframe writing to make the bigger picture (2) clearer. While the authors do not address 3, the scope of addressing this is incredibly large and is in my opinion more appropriate for future work. Finally, the authors correctly identify that NLP researchers creating vocabularies may lead to further inaccuracies and limit opportunities for interdisciplinary collaboration.